# OpenReview forum: "Laterality Failure in Vision-Language Models on Surface Clinical Photographs: A Pilot Safety Study"
_MIDL.io/2026/Short_Papers — MIDL 2026 - Short Papers Poster_

### Official Review · Reviewer_C235 · 2026-05-03
**Relevant question, good experimental setting despite small datasets, interesting insights**

**Rating:** 4
**Confidence:** 3

**Review:**

VLMs are compared against the task of assessing the laterality of photographed body parts. A two step process is followed to first select two most performing models, and then evaluate these two models under various prompt settings and image perturbations. Though very small datasets are used, general failure is observed and two different behaviors are exposed.

**Summary:**

The paper evaluates the capacity of vision language models to determine the laterality of a body part in a photography. 9 VLM are first evaluated on 8 images, then ChatGPT5.2 and MedGemma 4B are selected to be further evaluated on 10 images using prompting and image transform variations. The conclusion is a general lack of ability to consistently assess laterality (ChatGPT 5.2) or a bias in laterality assessment (MedGemma 4B), resulting in a general lack of trust in laterality assessed by VLMs.

**Strengths:**

- The addressed question is relevant in clinical practice.
- Though it has not been previously addressed as is, the authors relate their work to adequate previous investigation concerning spatial reasoning and VLMs
- Experiments are nicely devised and thereafter results are convincing

**Weaknesses:**

- The experiments rely on very small datasets, which hinders the explanatory analysis
- The authors made an overuse of supplementary material. At least some illustrations should have been kept in the main text

**Justification Of Rating:**

Relevant question, good experimental setting despite small datasets, interesting insights

---

### Decision · Program_Chairs · 2026-05-08

Accept (Poster)